# Periodontal Behavior Around Teeth Prepared with Finishing Line for Restoration with Fixed Prostheses. A Systematic Review and Meta-Analysis

**DOI:** 10.3390/jcm9010249

**Published:** 2020-01-17

**Authors:** Raquel León-Martínez, Jose María Montiel-Company, Carlos Bellot-Arcís, María Fernanda Solá-Ruíz, Eduardo Selva-Otaolaurruchi, Rubén Agustín-Panadero

**Affiliations:** Department of Dental Medicine Faculty of Medicine and Dentistry, University of Valencia, C/Gascó Oliag n1, Valencia 46010, Spain; raquelleonmtnz@gmail.com (R.L.-M.); jose.maria.montiel@uv.es (J.M.M.-C.); Carlos.Bellot@uv.es (C.B.-A.); eduardo.j.selva@uv.es (E.S.-O.); rubenagustinpanadero@gmail.com (R.A.-P.)

**Keywords:** periodontal prosthesis, biological factor, dental prosthesis, metal ceramic restorations, zirconium oxide

## Abstract

Background: The objective of this systematic review and meta-analysis was to analyze the periodontal behavior around teeth prepared with horizontal finishing crowns supporting fixed metal-ceramic and zirconia full coverage crowns and fixed partial dentures (FDPs). Materials and methods: An electronic search was conducted to locate relevant clinical trials in four databases: PubMed, Embase, Cochrane, and Scopus. A manual search was made in the reference sections of the articles identified for any additional articles. No restrictions were applied regarding year of publication or language. The following variables were considered in quantitative and qualitative analysis: probing pocket depth (PPD); probing attachment level (PAL); plaque control record (PCR); bleeding on probing (BOP); and gingival margin migration. Results: Twenty articles were selected for qualitative synthesis, and of these, nine underwent meta-analysis. Higher PCR was found in control teeth, while BOP, PPD, and PAL were higher around teeth prepared with horizontal finishing lines supporting complete coverage crowns/FDPs Gingival migration results were the clearest manifestation of compromised periodontal health around teeth prepared with horizontal finishing lines. Conclusions: Meta-analysis revealed that teeth prepared with horizontal finishing lines supporting crowns and FDPs present more periodontal disorders than untreated control teeth.

## 1. Introduction

Fixed tooth-supported restorations are a type of treatment that has been well documented over the years. Restorations fabricated with a metal core and a ceramic coating are considered the ‘gold standard’, although in recent years completely ceramic restorations have grown in popularity due to their esthetic properties, biocompatibility, and good mechanical behavior [1].

With this type of treatment, one of the main objectives is to achieve marginal stability as often gingival recession exposes the tooth-prosthesis termination line, a problem that may derive from the patient’s biotype, iatrogenic damage in tooth preparation, the position of the dental finishing line, chronic inflammation, or trauma caused by the patient (for example during tooth brushing) [2,3].

Three types of dental preparation have been documented: horizontal (straight shoulder, beveled shoulder, curved chamfer, sloping chamfer), vertical (knife edge), and preparation without finishing line (biologically oriented preparation technique) introduced by Loi [3,4].

The position of the finishing line in relation to the gingival margin (subgingival; juxtagingival or supragingival) has great influence on periodontal behavior around teeth supporting prosthetic restorations. Localized gingival inflammation, increased plaque and gingival indices, increased probing depth, and gingival margin migration are associated with subgingival finishing lines [5,6].

Over the years, numerous clinical trials have analyzed the mechanical behavior of tooth supported fixed prostheses, but few have investigated the periodontal responses around them [2,7,8,9,10,11,12]. In this context, the aim of this systematic review and meta-analysis was to analyze the periodontal behavior around teeth prepared with horizontal finishing lines supporting metal-ceramic and zirconia complete coverage crowns and fixed partial dentures (FDPs).

## 2. Materials and Methods

This bibliographic search was conducted following PRISMA (Preferred Reporting Items for Systemic Reviews and Meta-Analyses http://www.prisma-statement.org) guidelines for systematic reviews and meta-analyses. The review also fulfilled the PRISMA 2009 Checklist [13] and was registered in the PRISMA database (PROSPERO), registration number: CRD42019119185.

The PICO (population, intervention, comparison, outcome) question was: ‘What periodontal behavior can be expected around teeth prepared with horizontal finishing lines supporting metal-ceramic and zirconia crowns and bridges?’ with the following components: population: patients treated with metal-ceramic or zirconia full coverage crowns or FDPs; intervention: dental preparation with horizontal finishing line; comparison: untreated teeth (control) compared with teeth prepared with horizontal finishing line and outcome: periodontal behavior.

An electronic search was conducted in the following databases: PubMed; Scopus; Embase, and Cochrane. The search covered all the literature published internationally up to November 2018. The search included seven medical subject heading (MeSH) terms: ‘periodontal’; ‘biological agents’; ‘dental crown’; ‘fixed partial denture’; ‘zirconia’; ‘metal-ceramic’. The Boolean operators applied were (‘OR’ and ‘AND’), as well as (‘NOT’). The search terms were structured as follows: [(‘periodontal’) OR (‘biological agents’)] AND [(‘dental crown’) OR (‘fixed partial denture’)] AND [(‘Zirconia’) OR (‘metal-ceramic’)] NOT implant, together with the search filter (‘clinical trial’).

Two researchers (R.L.M.; R.A.P.) conducted the database searches in duplicate independently. Titles and abstracts were selected applying inclusion and exclusion criteria. One researcher (R.L.M.) extracted data on the relevant variables. The systematic review was carried out by (R.L.M.) and subsequent meta-analysis was performed by two researchers not involved in the selection process (C.B.A.; J.M.C.).

Inclusion criteria: Studies recorded in databases as prospective and retrospective randomized clinical trials (RCTs). Samples of patients aged 18 years old or over; patients treated with metal-ceramic crowns and/or FDPs, monolithic zirconia crowns and/or FDPs, or zirconia crowns and FDPs with feldspathic ceramic coating; follow-up period of at least 6 months. No restriction was placed on the year of publication or language. Exclusion criteria: systematic literature reviews, clinical cases, case series, and editorials; studies including patients under the age of 18 years old; studies with samples of five or fewer patients; studies including patients with previous periodontal pathology.

The following data were extracted from each article: author and year of publication; title and journal in which article was published; sample size (n); follow-up time; periodontal complications: probing pocket depth (PPD), probing attachment level (PAL), plaque control record (PCR), bleeding on probing (BOP), and gingival margin migration.

The risk of bias in the studies selected for review was assessed using two scales for methodological quality assessment of clinical trials: the PEDro scale and the Jadad scale. The PEDro scale consists of 11 items (each evaluated as present or absent) making a score of 0–10. Studies scoring 5 or over are classified as high quality, and at low risk of bias [14,15]. The Jadad scale consists of five items that evaluate randomization, researcher and patient blinding, and description of losses during follow-up producing a score of 0–5; scores of less than 3 are considered low quality [16].

The data included in meta-analysis were combined by means of random effects models expressed as forest plots. Heterogeneity was determined applying the Q and *I*^2^ tests; heterogeneity was considered to exist when the *p*-value was <0.1. When *I*^2^ results were between 25% and 50%, heterogeneity was considered slight, between 50% and 75% moderate, and >75% high. Effect size was estimated from mean values for probing pocket depth (PPD), probing attachment level (PAL), plaque control record (PCR), and bleeding on probing (BOP), as well as the percentage of samples suffering gingival margin migration.

Differences in mean values for all treatment groups and control groups (untreated) were calculated for each variable: plaque control record (PCR), bleeding on probing (BOP), and probing pocket depth (PPD), excluding probing attachment level (PAL) and percentage of samples undergoing gingival margin migration. Probing attachment level (PAL) and probing pocket depth (PPD) were evaluated by means of a maximum likelihood meta-regression random effects models generated for these variables over the follow-up period.

Publication bias was assessed by means of funnel plots and Duval and Tweedie’s trim and fill method which makes an estimation on the basis of imputed studies, evaluating the difference from the estimation obtained by observed studies in meta-analysis.

## 3. Results

The initial electronic search identified 335 articles in PubMed, 120 in Cochrane, 76 in Embase, 181 in Scopus, and three in grey literature. Of the total 715 works, 379 were discarded as duplicates.

After reading the titles and abstracts, a further 313 were eliminated leaving a total of 23. A further three were rejected as they failed to fulfill the following inclusion criteria: they did not include periodontal data, did not use in vivo patients, or used restorations fabricated from materials other than metal-ceramic or zirconia.

A final total of 20 articles were included in qualitative synthesis. Nine works were included in quantitative synthesis as these included all the data and variables required (Figure 1).

The results of methodological quality assessment using the Jadad and PEDro scales are shown in Table 1 and Table 2.

The Jadad scale obtained five articles with scores of <3 (low quality). In general, the criterion that failed to be fulfilled with most frequency was double blinding, only applied in seven of the works. The PEDro scale obtained 14 articles with scores of >5 indicating high methodological quality. Again, quality was most frequently compromised by failure to fulfill items related to subject, treatment, or measurement blinding. Both scales have shown a high correlation in the scores obtained (Pearson = 0.955).

Qualitative synthesis included 20 articles (Table 3). Sample sizes in the studies analyzed varied from 5 to 240 patients, with subject ages ranging from 23 to 70 years, and the number of restorations placed from 12 to 480. All the data analyzed pointed to a higher gingival index in treated teeth but lower amounts of plaque than untreated control teeth, also a higher pocket depth and less attachment level around the abutments.

As for the gingival margin migration variable, this result was estimated as a percentage of supporting teeth that underwent some change in the position of the gingival margin in relation to the finishing line. Valderhaug (1976) reports migration of the gingival margin around prepared teeth supporting fixed prostheses in 36.7% of samples [2]. In 1993, the same author observed migration in 43.7% of cases, while Peláez (2012) observed gingival margin migration in 77.7% of zirconia crowns and 16.6% of metal-ceramic crowns [5,11].

Qualitative analysis found that a plaque control record (PCR) had been employed to estimate plaque index in nine of the works included in meta-analysis; these data were combined in a random effects model, which estimated a mean plaque index of 0.25 (CI-95% 0.17–0.33). Q and *I*^2^ tests determined heterogeneity among the studies (Q = 47.8; *p*-value = 0.000; *I*^2^ = 83.3%). The funnel plot (Figure 2) shows some asymmetry. Adjusted estimation for the plaque index using Duval and Tweedie’s trim and fill method was 0.13, a value not included in the estimated confidence interval, indicating the existence of publication bias.

The difference in estimated mean plaque indices between prosthesis-supporting teeth and untreated control teeth was 0.14 points (CI 95% 0.07–0.21), significantly higher (Z = 6.11; *p* = 0.000) in the control group (Figure 3). The three studies included in meta-analysis had follow-up periods ranging from 40 to 60 months. Meta-analysis did not find heterogeneity (Q = 2.41; *p* = 0.299; *I*^2^ = 17.1%). Introducing imputed studies, trim and fill estimation was 0.10, not indicative of publication bias.

As for bleeding on probing (BOP), again data were combined using a random effects model of nine studies, obtaining a mean gingival index of 0.43 (IC−95% 0.30–0.55). Meta-analysis found high heterogeneity (Q = 58.8; *p* = 0.000; *I*^2^ = 86.4%). The funnel plot shows an asymmetrical image. Adjusted estimation for mean gingival index using Duval and Tweedie’s trim and fill method was 0.28, a value not included in the estimated confidence interval, indicating the existence of publication bias (Figure 4).

The difference in estimated mean gingival indices between treatment groups and control groups (Figure 5) was 0.04 points (IC 95% −0.03 and 0.14) without statistically significant difference (Z = 1.29; *p* = 0.196) between the two groups. The three articles included had follow-up periods ranging from 36 to 48 months. Meta-analysis did not observe heterogeneity (Q = 2.58; *p* = 0.275; *I*^2^ = 22.4%). When imputed studies were introduced using the trim and fill method, the difference between estimated means did not point to publication bias.

Regarding periodontal probing depth (PPD), a mean probing depth of 2.20 mm was estimated (IC-95% 2.01–2.39), combining 14 articles (as in one article variables differed in terms of time and materials) in a random effects model. Meta-analysis identified high heterogeneity among the works (Q = 610.5; *p*-value = 0.000; *I*^2^ = 97.8%). Duval and Tweedie’s trim and fill method did not produce evidence of publication bias. The funnel plot (Figure 6) showed symmetry.

Difference between mean periodontal probing depths in the two groups was found to be statistically significant (Z = 1.975; *p* = 0.045), estimated to be 0.11 mm (IC 95% 0.001–0.216) greater in treated teeth in comparison with untreated teeth (Figure 7). The three studies included had follow-up periods ranging from 40 to 180 months. Meta-analysis found only slight heterogeneity (Q = 2.99; *p* = 0.224; *I*^2^ = 33.2%). When imputed studies were introduced, estimation did not differ from observed studies, pointing to no publication bias.

When probing depth was analyzed by random effects meta-regression (Figure 8) and maximum likelihood, with mean probing depth as dependent variable and follow-up time as co-variable, the model did not obtain significance (Q = 1.38; *p* = 0.240; *R*^2^ = 0.03). Follow-up duration did not influence (*p* = 0.240) probing depth (beta coefficient = 0.005; 95% CI −0.003; 0.014).

As for probing attachment level (PAL), a random effects model of five studies estimated mean PAL of 1.61 mm (IC 95% 0.66–2.56). Meta-analysis showed high heterogeneity among the studies (Q = 1626.4; *p*-value = 0.000; *I*^2^ = 99.7%). Meta-analysis did not point to publication bias, presenting a symmetrical funnel plot and no difference between observed and adjusted estimations according to Duval and Tweedie’s trim and fill method (Figure 9).

When probing, attachment level was analyzed by random effects meta-regression (Figure 10) and maximum likelihood, with mean probing attachment level as dependent variable and follow-up time as co-variable, the model obtained was significant (Q = 319.5; *p* = 0.000; R^2^ = 0.99), with an intercept –0.926 (−1.235; −0.618). The follow-up variable was found to be highly significant and predictive (*p* = 0.000) obtaining a beta coefficient = 0.090 (0.079; 0.098), indicating a loss of 0.09 mm insertion level for each month passed.

Lastly, four studies were included in a random effects model of gingival margin migration (Figure 11), obtaining a value of 40.7% (IC 95% 30.7%–51.6%). The heterogeneity detected was moderate (Q = 9.1; *p*-value = 0.028; *I*^2^ = 66.9%). The funnel plot showed some symmetry. Adjusted estimation using Duval and Tweedie’s trim and fill method for gingival margin migration was 37%, a percentage included in the estimated confidence interval, indicating no publication bias.

## 4. Discussion

When it comes to dental preparation before the subsequent placement of a crown or FDP, the decisions taken are mainly based on the type of restoration material or the characteristics of the tooth requiring restoration. It is also important to take the surrounding periodontal tissues into account, as these may suffer iatrogenic damage if dental preparation is not performed correctly. In this context, awareness of periodontal pathology derived from dental preparation and fixed prostheses is very relevant and is the subject of this systematic review and meta-analysis.

Most research and meta-analyses of published literature have focused on the prostheses’ mechanical complications, for example chipping, and have placed periodontal complications in the background. [10,11,12,17,18,19,20,21,22,23,24,25,26,27]. Only a few works have investigated gingival and periodontal health around crowns and FDPs exclusively [2,5,7,8,20,22]. For this reason, the present meta-analysis examined periodontal complications around teeth prepared with horizontal finishing lines supporting fixed prosthetic restorations.

This systematic review followed Preferred Reporting Items for Systematic Reviews and Meta-Analyses (PRISMA) guidelines in both the review and subsequent meta-analysis. Based on the PICO question established at the outset, a search was conducted in the PubMed, Cochrane, Embase, and Scopus databases, as well as in grey literature, without placing any limitation on the year of publication or language, applying strict inclusion and exclusion criteria in the selection process. Qualitative analysis of the most relevant data was performed (gingival margin migration, insertion loss, periodontal probing depth, plaque index, and gingival index), followed by quantitative analysis performed systematically following methods established in previous works, in particular, analyzing heterogeneity with the Q and *I*^2^ tests, and publication bias by means of funnel plots [28,29,30]. The studies classified as low quality using 2 scales such as Jadad and PEDro have not been included in the different meta-analysis performed.

In the studies selected for analysis, gingival index, plaque index, probing depth, gingival margin migration and insertion loss were evaluated as indications of whether or not periodontal complications were present.

A higher gingival index was found in teeth prepared with horizontal finishing line than untreated control teeth. As for periodontal probing depth, several articles reported significantly greater depth observed around restored teeth than untreated control teeth [18,19,20]. Suárez and Sax found greater insertion loss in teeth prepared with horizontal finishing line than untreated teeth [12,13,14,15,16,17,18,19,20,21,22,23,24,25,26,27]. Lastly, the parameter that most clearly pointed to periodontal disorders was gingival margin migration as reported in studies such as those by Peláez or Valderhaug [2,5,11]. Gingival migration is due partly to individual patient factors such as oral hygiene or periodontal treatment carried out prior to treatment [2,5,8,11,12,20]. According to various authors, the main reason for increased gingival bleeding and periodontal inflammation is poor marginal adaptation between the restoration and the prepared tooth [8,11,12,20,24]. In this context, the position of the margin is important, as subgingival dental preparation has been seen to affect periodontal health negatively [2,5,7,8,11,21,24].

At the same time, the plaque index was lower around teeth restored with complete coverage crowns, as ceramic materials have been shown to retain less plaque than natural teeth [8,24,25]. In articles that evaluated periodontal parameters in relation to the restoration material used, different levels of gingival bleeding were observed, which were sometimes contradictory. A clinical trial by Suárez found better results around zirconia crowns and FDPs, while Sailer observed more bleeding around zirconia restorations [9,10,11].

According to Tanner and Sailer, this is due to the difficulty of hygiene maintenance around FDPs derived from the presence of connectors of large dimensions that may compromise interproximal spaces [9,10,24]. In the study by Peláez, who compared gingival margin migration, 77.7% of zirconia crowns were affected compared with 16.6% of metal-ceramic crowns, although no conclusion was drawn as to the cause of this difference [11]. For this reason, more randomized clinical trials are needed to investigate periodontal responses to restorations fabricated from these materials.

The variables that showed the greatest relevance in the present meta-analysis, providing the clearest indications of periodontal status around crowns and FDPs on teeth prepared with horizontal finishing lines were periodontal probing depth, insertion loss, and gingival margin migration. Regarding periodontal probing depth, the studies analyzed showed asymmetry and a mean of around 2.2 mm which, although this is not considered unhealthy, was nevertheless greater than that of untreated control teeth [2,5,20,24]. As for insertion loss, analysis obtained a mean of 1.61 mm, which is considered slight, but still higher than for untreated teeth [31]. When meta-regression was performed for probing attachment level, this variable was related to time, with greater loss of attachment in studies with longer follow-up periods. However, when meta-regression was applied to probing depth, this variable did not present the same relation with time, a coherent finding given that loss of insertion is related to apical migration of the gingival margin and so is not accompanied by an increase in probing depth. The variable that was most indicative of anomalous periodontal behavior around restorations supported by teeth prepared with horizontal finishing line was gingival margin migration. The studies subjected to meta-analysis showed that 40.7% of samples suffered migration, in other words, gingival recession was more prevalent in crowns and FDPs placed subgingivally [2,5,11,30,32].

## 5. Conclusions

Within the limitations of this systematic review and meta-analysis, the following conclusions may be drawn:Crowns and FDPs on teeth prepared with horizontal finishing line present poorer periodontal health in all the periodontal variables analyzed compared with untreated control teeth.No conclusive relations can be established between periodontal behavior and the materials used to fabricate crowns and FDPs.

## Figures and Tables

**Figure 1 jcm-09-00249-f001:**
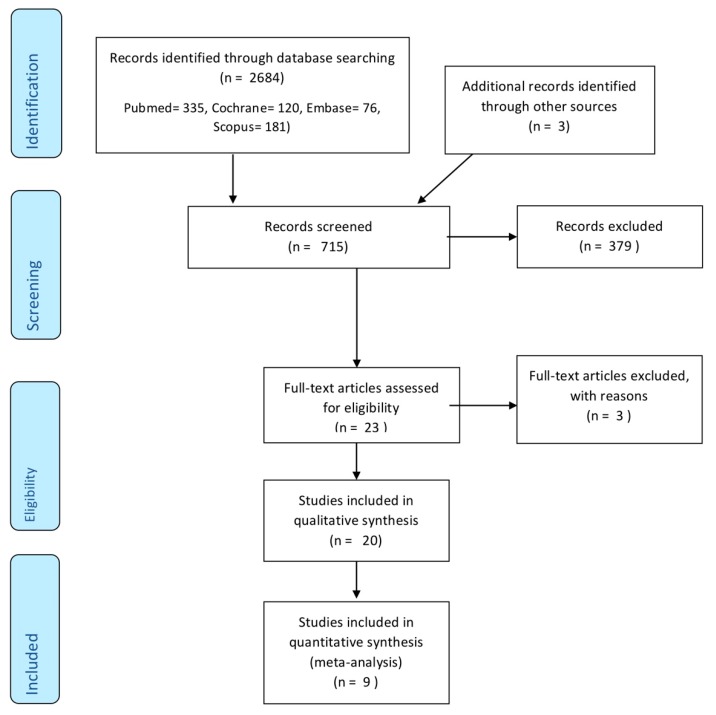
Flow chat of study selection procedure.

**Figure 2 jcm-09-00249-f002:**
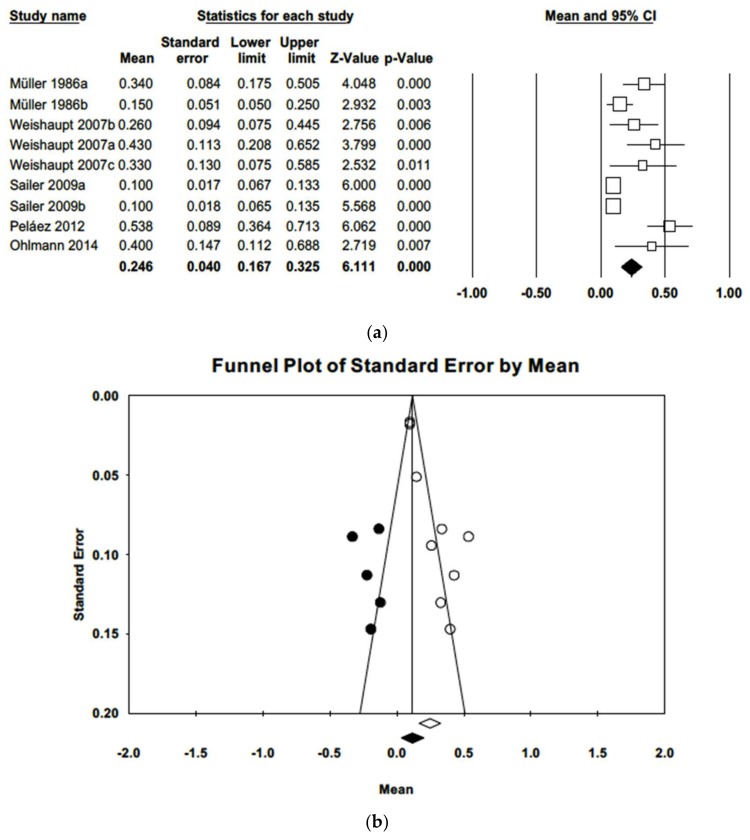
(**a**) Forest plot of plaque index meta-analysis with observed and imputed studies. (**b**) Funnel plot of plaque index meta-analysis with observed and imputed studies.

**Figure 3 jcm-09-00249-f003:**
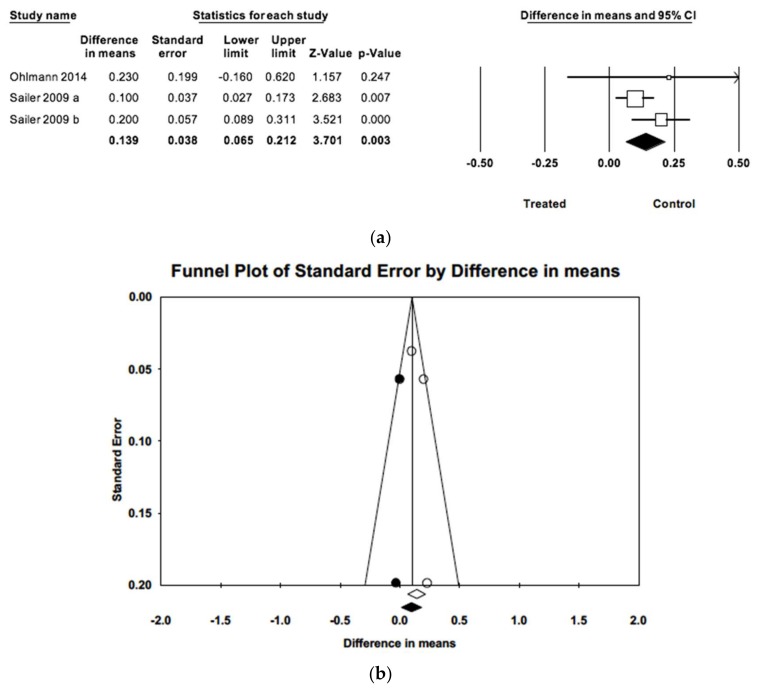
(**a**) Forest plot of meta-analysis of the difference in mean gingival indices between treated and control samples. (**b**) Funnel plot of observed and imputed studies.

**Figure 4 jcm-09-00249-f004:**
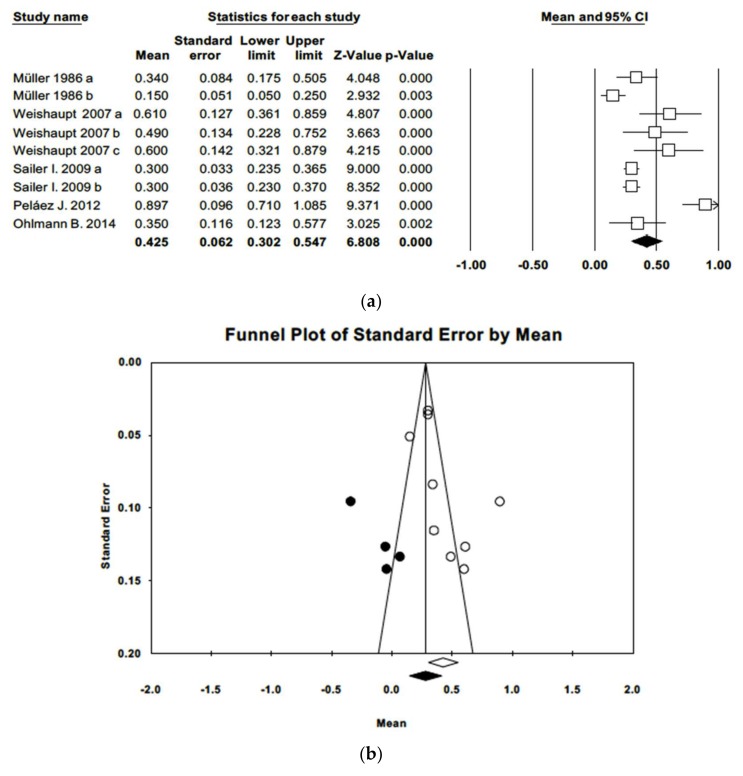
(**a**) Forest plot of gingival index meta-analysis. (**b**) Funnel plot of gingival index with observed and imputed studies.

**Figure 5 jcm-09-00249-f005:**
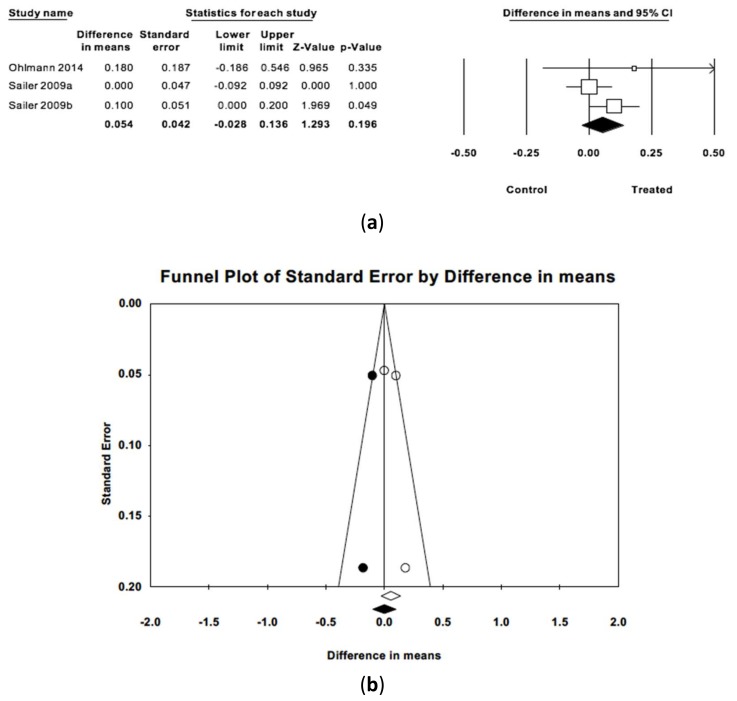
(**a**) Forest plot of meta-analysis of the difference in mean gingival indices between treated and control groups. (**b**) Funnel plot with observed and imputed studies.

**Figure 6 jcm-09-00249-f006:**
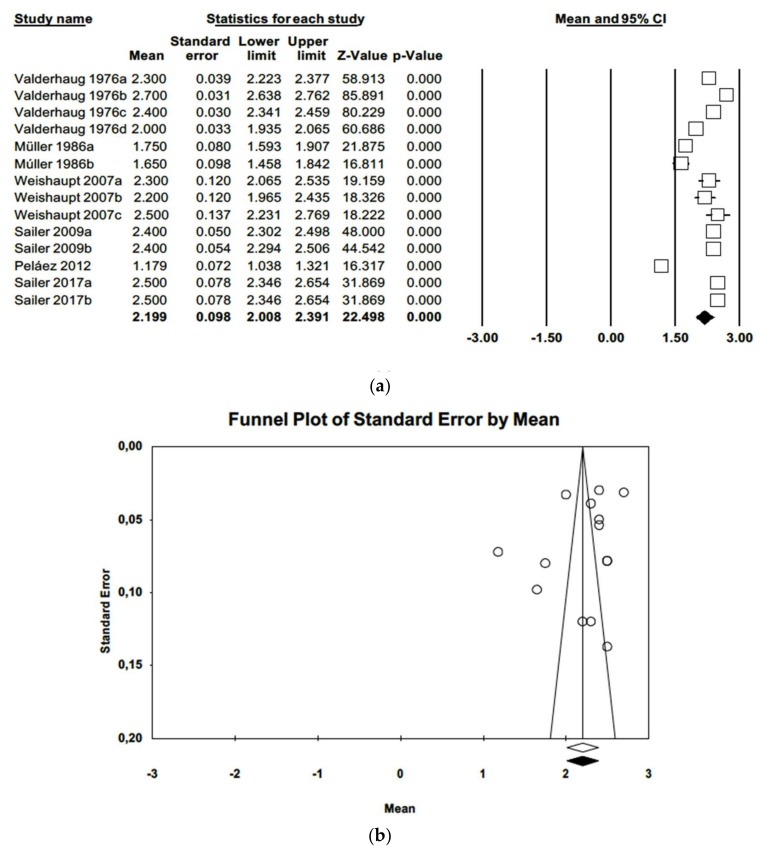
(**a**) Forest plot of periodontal probing depth meta-analysis. (**b**) Funnel plot with observed and imputed studies.

**Figure 7 jcm-09-00249-f007:**
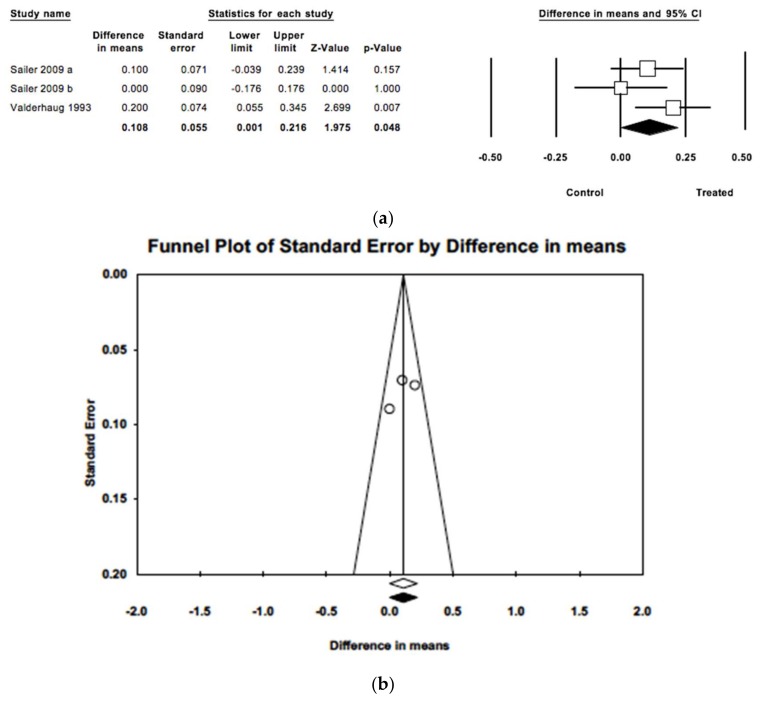
(**a**) Forest plot of meta-analysis of mean probing depths comparing treated and untreated groups. (**b**) Funnel plot with imputed and observed studies.

**Figure 8 jcm-09-00249-f008:**
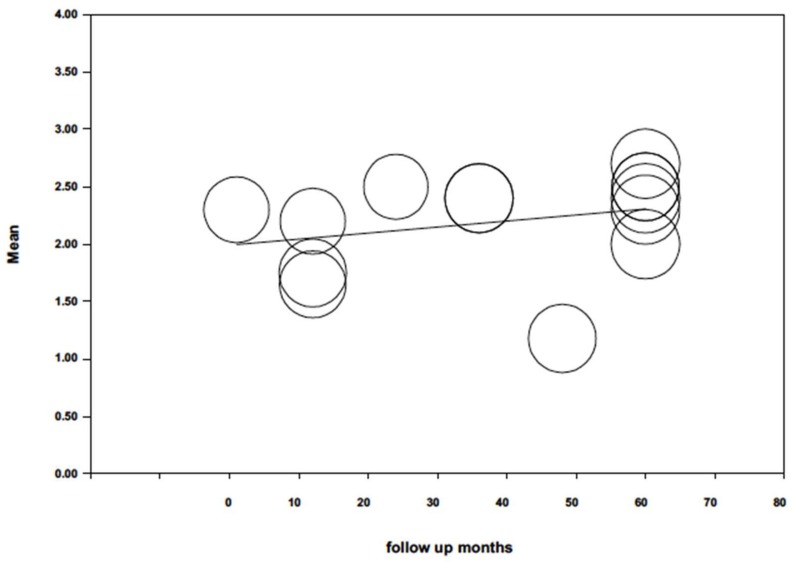
Scatter plot of mean periodontal probing depth meta-regression over time.

**Figure 9 jcm-09-00249-f009:**
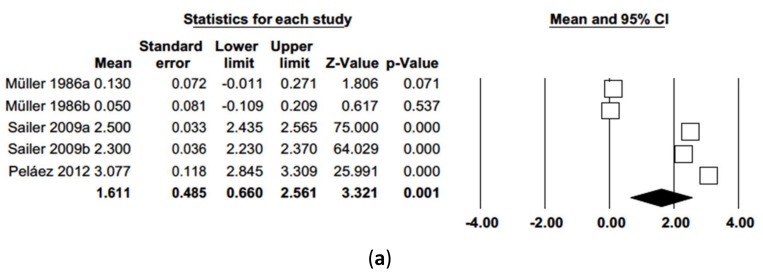
(**a**) Forest plot of probing attachment level meta-analysis. (**b**) Funnel plot with observed and imputed studies.

**Figure 10 jcm-09-00249-f010:**
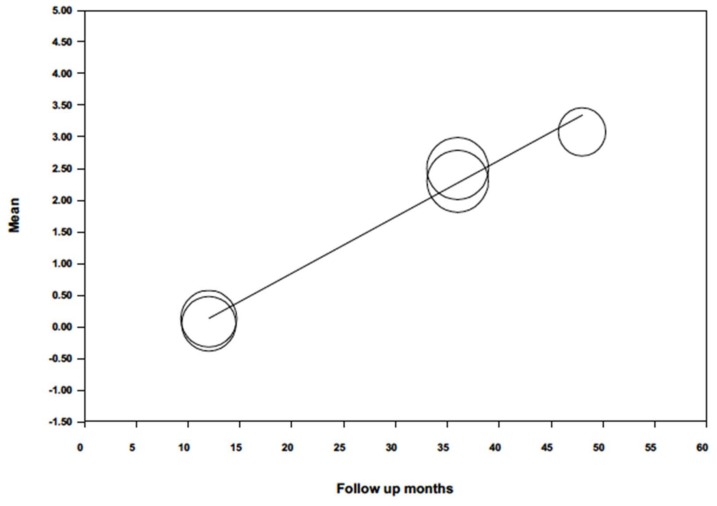
Scatter plot of mean probing attachment level meta-regression over time.

**Figure 11 jcm-09-00249-f011:**
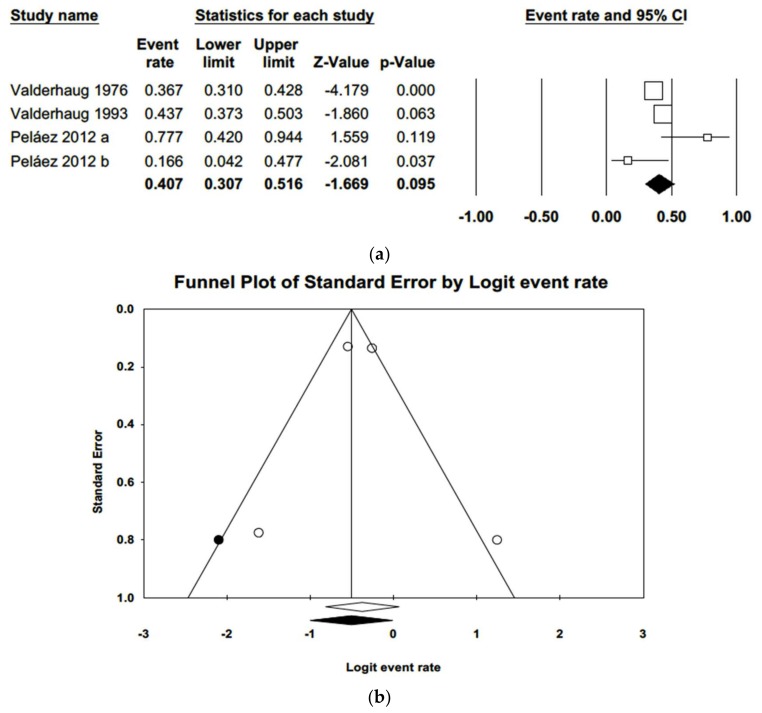
(**a**) Forest plot of meta-analysis % of samples suffering gingival margin migration. (**b**) Funnel plot with observed and imputed studies.

**Table 1 jcm-09-00249-t001:** Assessment of methodological quality according to the Jadad scale.

JADAD CRITERIA						
Author/Year	Is the Study Described as Randomized?	Is the Study Described as Double-Blinded?	Was There a Description of Withdrawals and Dropouts?	Was the Method of Randomization Adequate?	Was the Method of Blinding Appropriate?	Score
Sailer I. et al. 2017	1	0	1	1	0	3
Sailer I. et al. 2009	1	0	1	1	0	3
Molin M. et al. 2008	0	0	1	0	0	1
Zenthöfer A. et al. 2015	1	1	1	1	1	5
Reitemeier B. et al. 2002	1	0	1	1	0	3
Sax C. et al. 2011	0	0	1	0	0	1
Ioannidis A. et al. 2016	0	0	1	0	0	1
Suarez MJ. et al. 2018	1	1	1	1	1	5
Peláez J. et al. Oct 2012	1	1	1	1	1	5
Peláez J. et al. Jun 2012	0	0	1	0	0	1
Nicolaisen MH. et al. 2016	1	1	1	1	1	5
Setz J. et al. 1994	1	1	1	1	1	5
Tanner J. et al. 2018	0	0	1	0	0	1
Hἀff A. et al. 2014	0	0	1	0	0	1
Ohlmann B. et al. 2014	1	0	1	1	0	3
Naenni N. et al. 2015	1	1	1	1	1	5
Weishaupt P. et al. 2007	1	1	1	1	1	5
Valderhaug J. et al. 1993	1	0	1	1	0	3
Valderhaug J. et al. 1976	1	0	1	1	0	3
Müller HP. et al. 1986	1	0	1	1	0	3

**Table 2 jcm-09-00249-t002:** Assessment of methodological quality according to PEDro scale.

CRITERIA PEDro												
Author/Year	Eligibility Criteria Were Specified	Subjects Were Randomly Allocated to Groups	Allocation Was Concealed	The Groups Were Similar at Baseline	There Was Blinding of All Subjects	There Was Blinding of All Therapists Who Administered the Therapy	There was BLINDING of All Assessors	Measures Were Obtained from More Than 85% of the Subjects	Results Obtained for All Subjects	The Results of between-Group Statistical Comparisons Are Reported	The Study Provides Both Point Measures and Measures of Variability	Score
Sailer I. et al. 2017	yes	yes	yes	yes	yes	no	no	yes	yes	yes	yes	8
Sailer I. et al. 2009	yes	yes	yes	yes	yes	no	no	yes	yes	yes	yes	8
Molin M. et al. 2008	yes	no	no	no	yes	no	no	yes	no	no	no	3
Zenthöfer A. et al. 2015	yes	yes	yes	yes	yes	no	yes	yes	yes	yes	yes	9
Reitemeier B. et al. 2002	yes	yes	no	no	yes	no	no	yes	yes	yes	yes	6
Sax C. et al. 2011	yes	no	no	no	yes	no	no	no	no	no	yes	3
Ioannidis A. et al. 2016	yes	no	no	no	yes	no	no	no	no	no	yes	3
Suarez MJ. et al. 2018	yes	yes	yes	yes	yes	no	yes	yes	yes	yes	yes	9
Peláez J. et al. Oct 2012	yes	yes	yes	yes	yes	no	yes	yes	yes	yes	yes	9
Peláez J. et al. Jun 2012	yes	no	no	no	yes	no	yes	no	no	no	yes	4
Nicolaisen MH. et al. 2016	yes	yes	yes	yes	yes	no	yes	yes	yes	yes	yes	9
Setz J. et al. 1994	yes	yes	yes	yes	yes	yes	yes	yes	yes	yes	yes	10
Tanner J. et al. 2018	yes	no	no	no	no	no	no	no	no	no	yes	2
Hἀff A. et al. 2014	yes	no	no	no	no	no	no	no	no	no	yes	2
Ohlmann B. et al. 2014	yes	yes	yes	yes	yes	no	no	yes	yes	yes	yes	8
Naenni N. et al. 2015	yes	yes	yes	yes	yes	yes	no	yes	yes	yes	yes	9
Weishaupt P. et al. 2007	yes	yes	yes	yes	yes	no	yes	yes	yes	yes	yes	9
Valderhaug J. et al. 1993	yes	no	no	no	yes	no	no	yes	yes	yes	yes	6
Valderhaug J. et al. 1976	yes	no	no	no	yes	no	no	yes	yes	yes	yes	6
Müller HP. et al. 1986	yes	no	no	no	yes	no	no	yes	yes	yes	yes	6

**Table 3 jcm-09-00249-t003:** Quantitative analysis of articles included in the systematic review.

Author/Year	Study Tipe	N	Sex/Mean Age/Losses	T (Months)	Periodontal	Complications			PEDro	Jadad
					PPD	PAL	PCR	BOP		
Sailer I. et al. 2017	RCT	62 −22 MC −40 ZR	22 f/24 m 56 y 7 l	60	ZR 2.5 ± 0.4 mm MC 2.6 ± 0.4 mm	ZR 0.0 ± 0.2 mm MC 0.0 ± 0.2 mm	ZR 13.8 ± 24.5% MC 12.9 ±17.8%	ZR 32.8 ± 26.7% MC 29.8 ± 24%	8	3
Sailer I. et al. 2009	RCT	58 −24 MC −34 ZR	22 f/25 m 54.4 ± 12.7 y 6 l	40	ZR 2.4 ± 0.3 mm MC 2.4 ± 0.3 mm	ZR 2.5 ± 0.2 mm MC 2.3 ± 0.2 mm	ZR: 0.1 ± 0.1 MC: 0.1 ± 0.1	ZR: 0.3 ± 0.2 MC: 0.3 ± 0.2	8	3
Molin M. et al. 2008	PCT	57 -Control (untreated teeth)	12 f/6 m 57 y 0 l	60	-	-	No differences regarding control group.	No differences regarding control group.	3	1
Zenthöfer A. et al. 2015	RCT	19 −10 ZR −9 MC	11 f/8 m 56 y 2 l	36	No differences regarding ZR or MC crowns.	No differences regarding ZR or MC crowns.	No differences regarding ZR or MC crowns.	No differences regarding ZR or MC crowns.	9	5
Reitemeier B. et al. 2002	PCT	480 -Control (untreated teeth)	160 f/80 m 42.3 ± 3 y 0 l	12	-	-	Greater amount of plaque in control teeth.	Crowned teeth subgingival double probability bleeding before supragingival. No differences regarding control group.	6	3
Sax C. et al. 2011	PCT	85 -Control (untreated teeth)	9 f/12 m 48.3 ± 10 y 16 l	120	-	Test 0.7 mm Control 0.46 mm	-	-	3	1
Ioannidis A. et al. 2016	PCT	171 -Control (Ramfjorf teeth)	31 f/22 m 52.6 ± 10 y 2 l	120	-	-	No differences.	Greater gingival bleeding in crowned teeth.	3	1
Suarez MJ. et al.2018	RCT	120 −60 MC −60 ZR -Control (untreated teeth)	23f/17 m 24–70 y 0 l	60	-	Greater amount of PAL in MC and ZR compared to the control group.	Greater amount of plaque in MC and ZR versus control group.	Greater gingival bleeding in MC with respect to ZR. Greater bleeding MC and ZR versus control teeth.	9	5
Peláez J et al. 2012	PCT	60 -Control (untreated teeth)	11 f/6 m 23–65 y 0 l	36	1.17 ± 0.45	3.07± 0.73	0.53 ± 0.55	0.89 ± 0.59	4	1
Nicolaisen MH. et al. 2016	RCT	102 51 MC 51 ZR	21f/12 m 51 y 0 l	36	Increase of 0.1 mm MC and 0.2 mm ZR (initial <4 mm)	Increase of 0.9 ± 1.2 mm MC (initial 1.8 mm) Increase of 0.7 ± 0.8 mm ZR (initial 1.7 mm)	-	-	9	5
Setz J. et al. 1994	PCT	12 -Control (untreated teeth)	-	6	Significant differences with higher PPD in treated teeth versus control.	-	-	-	10	5
Tanner J. et al. 2018	ReCT	40 -Control (untreated teeth)	19 f/8 m 64.6 y 8 l	67	23% > 5 mm test 5% > 5 mm control	-	Average plaque index 0.2 test 0.6 control.	38.1% test 13.9% control.	2	1
Hἀff A. et al. 2014	ReCT	118 -Control (dientes no tratados)	24f/6 m 68 ± 11 y 26 l	156	-	-	13% control 8.3% test	8.3% control 8.3% test	2	1
Ohlmann B. et al. 2014	RCT	111 37 MC (control) 36 PC 38 PC + F	37 f/29 m 46 y 9 l	48	-	-	MC 0.40 ± 0.75 PC 0.56 ± 0.73 PC + F 0.63 ± 0.76	MC 0.35 ± 0.59 PC 0.56 ± 0.81 PC + F 0.35 ± 0.59	8	3
Naenni N. et al. 2015	RCT	108 54 ZR 54 LD	25 f/11 m 52.3 y 4 l	36	No differences regarding groups.	-	Greater amount of plaque in LD.	No differences regarding groups.	9	5
Weishaupt P. et al. 2007	RCT	68	34 - 18 l	24	-	-	T1: 0.43 ± 0.66 T2: 0.6 ± 0,83	T1: 0.61 ± 0.74 T2: 0.60 ± 0.83	9	5
Peláez J. et al. 2012	PCT	120 60 MC 60 ZR	22 f/15 m 23–65 y 0 l	48	-	Marginal integrity: 85% Zirconia 75% MC	No differences regarding groups.	No differences regarding groups.	9	5
Valderhaug J. 1993	PCT	187	55 - 156 l	180	57% 2 mm 34% 4 o > mm	43.7% changes in level of insertion.	27% 15 years later.	63% subgingivals crowns.	6	3
Valdergaug J 1976	PCT	380	98 40–60 y 16 l	60	83% 2 mm	36.7% changes in level of insertion.	No differences.	69% subgingivals crowns.	6	3
Müller HP. 1986	PCT	47	5 - 0 l	12	Supragingival: 1.75 ± 0.4 mm Juxtagingival: 1.65 ± 0.34 12% 3 mm o >	Supragingival: +0.13 ± 0.36 Juxtagingival: −0.05 ± 0.38	Supragingival: 0.34 ± 0.42 Juxtagingival: 0.15± 0.24 74% sin placa	Supragingival: 0.17 ± 0.27 Juxtagingival: 0.60 ± 0.42 74% without bleeding.	6	3

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
