# Peer review of "Periodontal Behavior Around Teeth Prepared with Finishing Line for Restoration with Fixed Prostheses. A Systematic Review and Meta-Analysis"

_jcm, 2020, doi:10.3390/jcm9010249_

Round 1

Reviewer 1 Report

The authors present a systematic analysis on the peridontal behaviro around teeth prepared with horizontal finishing crowns.  20 articles were chosen for qualitative analysis. The authors reached a conclusion that teeth prepared with horizontal finishing lines 26 supporting crowns and FPDs present more periodontal disorders than untreated control teeth. 

The article was well written and present. My only suggestion is to clear up some of the figures as they are too small to visualize. Another concern is that whether this type of contribution to the journal should be categorized into review or a big-data research article. 

Author Response

I thank you for the suggestions you made. We have enlarged the size of the figures for perfect viewing.

We believe that the article fits to the “review” category better, as the instructions in the Authors guidelines of the magazine point out.

Reviewer 2 Report

I read with interest in your manuscript.

Please consider the following recommendations:

Provide the rationale for using both PEDro and Jadad scale and how their scores relate to each other. More information regarding how the quality of the included studies influenced the clinical recommendations is pertinent. Use the Glossary of the JPD to refer to the terms adequately (e.g., use fixed dental prosthesis [FDP] instead of fixed partial denture [FPD]). Remove Figure 1 (PRISMA checklist) from the manuscript and use it either for the editorial reviewers only or attach it as an Appendix.

Author Response

I appreciate the questions you ask us and that I address below.

JADAD and Pedro are two quality scales used in the evaluation of experimental studies. Jadad has more limitations than Pedro since it only uses 5 items while Pedro uses 9 and is more complete. Both scales have shown a high correlation with each other (Pearson = 0.955).

The  studies classified as low quality have not been included in the different meta-analyzes performed and therefore they have not affected the estimates obtained. We have added in the manuscript’s results: Both scales have shown a high correlation in the scores obtained (Pearson = 0.955). We have added in the manuscrpt’s discussion: The studies classified as low quality using 2 scales such as Jadad and PEDro have not been included in the different meta-analysis performed We have withdrawn Figure 1 as suggested and we have changed the term FPD to FDP following JPD’s glossary.